# Novel 1,2,3-Triazole Derivatives as Mimics of Steroidal System—Synthesis, Crystal Structures Determination, Hirshfeld Surfaces Analysis and Molecular Docking

**DOI:** 10.3390/molecules26134059

**Published:** 2021-07-02

**Authors:** Mateusz Daśko, Anna Dołęga, Magdalena Siedzielnik, Karol Biernacki, Olga Ciupak, Janusz Rachon, Sebastian Demkowicz

**Affiliations:** 1Department of Inorganic Chemistry, Faculty of Chemistry, Gdańsk University of Technology, Narutowicza 11/12, 80-233 Gdańsk, Poland; anna.dolega@pg.edu.pl (A.D.); m.siedzielnik@gmail.com (M.S.); 2Department of Organic Chemistry, Faculty of Chemistry, Gdańsk University of Technology, Narutowicza 11/12, 80-233 Gdańsk, Poland; karol.biernacki@pg.edu.pl (K.B.); olgaciupak@gmail.com (O.C.); januszrachon@gmail.com (J.R.); sebastian.demkowicz@pg.edu.pl (S.D.)

**Keywords:** triazoles, hormone analogs, drug design, crystal structures, Hirshfeld surface, molecular docking

## Abstract

Herein, we present the synthesis and crystal structures determination of five 4-(1-phenyl-1*H*-1,2,3-triazol-4-yl)phenol derivatives containing halogen atoms, **6a**–**e**, which may be used as an excellent mimic of steroids in the drug development process. Good quality crystals obtained for all of the synthesized compounds allowed the analysis of their molecular structures. Subsequently, the determined crystal structures were used to calculate the Hirshfeld surfaces for each of the synthesized compounds. Furthermore, results of our docking studies indicated that synthesized derivatives are able to bind effectively to the active sites of selected enzymes and receptors involved in the hormone biosynthesis and signaling pathways, analogously to the native steroids.

## 1. Introduction

Triazoles are a class of compounds showing very interesting properties, e.g., hydrogen bond (HB) formation, π–π stacking interaction, large dipole moments, bioisosteric effects, and therefore, they have been successfully used as scaffolds in the synthesis of antimicrobial, antiviral, and antitumor agents [1]. Importantly, they do not undergo hydrolysis under acidic or basic conditions, and they withstand metabolic degradation, which is desired in the design of new pharmaceuticals. The above advantages indicate that the derivatization of the triazole ring may lead to compounds possessing interesting biological properties. For example, the introduction of two additional phenyl rings with diverse substituents to the 1,2,3-triazole ring should allow the production of 4-(1-phenyl-1*H*-1,2,3-triazol-4-yl)phenol derivatives demonstrating similarities to some natural compounds, e.g., hormones.

The hormone signaling pathway is a well-established target for the development of hormone-dependent cancer drugs (e.g., breast cancer) [2]. For example, one of the clinically used drugs—*Tamoxifen* **1** (Figure 1) acts as a selective estrogen receptor modulator (SERM). On the other hand, chemotherapeutics, which may influence the hormone formation process, are of high therapeutic importance. The biosynthesis of active steroids (e.g., estradiol (E2) and androstenediol (Adiol)) in cancer tissues mainly depends on three enzymatic pathways: aromatase (AROM), 17β-hydroxysteroid dehydrogenase (17β-HSD) and steroid sulfatase (STS) [3]. For example, currently used *Letrozole* **2** and *Anastrozole* **3** (Figure 1) block the conversion of androgens to estrogens via the inhibition of the AROM complex. In light of recent research indicating that the disorders in sulfation/desulfation processes may be responsible for numerous pathologies [4], the other enzyme implicated in the steroidogenesis process—STS—is becoming a new, interesting molecular target in the development of novel and effective hormone-dependent cancer treatment methods. Recently, we have developed a series of tricyclic compounds, featuring the 1,2,3-triazole unit, as potent STS inhibitors [5,6]. In the course of our investigation, we have found that the most active analog, *MD77* **4** (Figure 1), inhibited the STS enzyme with an IC_50_ value of 36 nM when evaluated in an enzymatic assay. Our studies indicated that the highest inhibitory activities were exhibited by derivatives containing a fluorine atom at the meta position of the terminal aromatic ring. According to the molecular docking calculations, it was noticed that the fluorine atoms presented in *MD77* may interact with the Arg98 residue located in the STS active site. This additional interaction may stabilize the inhibitor–enzyme complex, resulting in improved inhibitory activity [5].

In the present paper, we have reported the crystal structures and Hirshfeld surfaces determination of five 4-(1-phenyl-1*H*-1,2,3-triazol-4-yl)phenol derivatives. The crystal structures provide insights into the possible conformations and the character of the intermolecular interactions of different derivatives. Additionally, we have described preliminary docking studies indicating effective binding of the synthesized compounds to the active sites of AROM, STS, 17β-HSD, and estrogen receptors (ER). The collected data indicate that the reported triazole derivatives could be utilized in the development of novel inhibitors of proteins involved in the hormone biosynthesis and signaling pathways and thus become a perfect starting point in the development of novel anticancer agents.

## 2. Results and Discussion

### 2.1. Chemistry

Three compounds **6a**–**c** were resynthesized according to the two-step synthetic protocol that we previously described (Scheme 1) [5]. Two novel compounds **6d** and **6e** were synthesized according to the same synthetic pathway using chlorinated aniline derivatives. In the first step, 4-[(trimethylsilyl)ethynyl]phenol **5** was obtained by the Sonogashira reaction between the 4-iodophenol and ethynyltrimethylsilane in the presence of palladium(II) chloride (PdCl_2_), triphenyl phosphine (Ph_3_P), copper(I) iodide (CuI) and triethylamine (NEt_3_). The Sonogashira coupling is a widely used cross-coupling reaction applied in organic practice to generate carbon–carbon bonds [7]. Next, the appropriate aniline derivatives were transformed into the corresponding azides with *tert*-butyl nitrite (*t*-BuONO) and azidotrimethylsilane (TMSN_3_). Then, to the obtained solution, 4-[(trimethylsilyl)ethynyl]phenol **5**, a 1 M solution of tetrabutylammonium fluoride (TBAF) in tetrahydrofuran (THF), copper(II) sulfate pentahydrate (CuSO_4_·5H_2_O) and a 1 M aqueous solution of sodium ascorbate were added. After a brief work-up of the reaction mixture, the corresponding 4-(1-phenyl-1*H*-1,2,3-triazol-4-yl)phenol derivatives **6a**–**e** were isolated. Recrystallization in acetonitrile (ACN) (**6c**–**e**), methanol (**6b**), or acetone (**6a**) allowed to obtain crystals of compounds suitable for X-ray diffraction measurement.

### 2.2. Crystal Structures and Hirshfeld Surfaces Analysis

Experimental X-ray crystal structures of **6a**–**e** may be helpful in the identification of the intermolecular interactions that can arise between the studied molecules and their environment. Good quality crystals obtained for all of the compounds allowed the analysis of their molecular structures (for the parameters of crystallographic data, see Table 3 in the Materials and Methods Section). The asymmetric units contained either one or two molecules of **6a**–**e**; typical examples and numbering schemes are illustrated in Figure 2a–c for compounds **6a**, **6c**, and **6d**, respectively (the molecular structures of the remaining two compounds can be found as Appendix A in *ESI*). Bond lengths between the atoms, which are collected in Appendix A in *ESI,* were very similar in all molecules and, in our opinion, did not depend much on the substitution pattern; the mean values of C–C bond lengths in the phenol rings were very close when compared between the derivatives ranging from 1.390 to 1.393 Å. The mean values of C–C bonds in the halogenated rings were invariably shorter and varied from 1.380 Å to 1.387 Å (Appendix A, *ESI*).

The substituted phenyl rings in **6a**–**e** can adopt different mutual orientations. The torsion angles between the six-membered rings varied from 0.90° in **6e** to 50.41° in **6a**. It can be observed that the same molecule can exhibit different torsion angles within the same crystal—as in **6a** and **6e** (Appendix A, *ESI*). Moreover, the mutual rotation of the rings may introduce a helical type of chirality, and we do observe two helical enantiomers of **6a** in crystals. In the examples presented in Figure 2 for **6a**, **6c**, and **6d** compounds, atoms of halogens are asymmetrically attached to the phenyl ring (with regard to triazole substitution), which leads to two possibilities of their orientation with regard to the triazole ring. The mono-substituted rings in **6a** and **6d** realize common conformation with the torsions N2-N3-C11-F1/Cl1 17.46°/20.32°, respectively, whereas in **6c,** the additional intramolecular contact between H8 and F1 (2.415 Å) may impose the opposite arrangement with the torsion N2-N3-C11-F2 equal to 154.62° (Figure 2).

Geometrical parameters of intermolecular interactions characteristic for compounds **6a**–**e** are collected in Appendix A (*ESI*) and the crystal packings are illustrated in Figure 3a,b and in Appendix A (*ESI*). The most typical interactions between the molecules included classical hydrogen bonds, which usually formed between the phenol group of one molecule and N1/N2 atoms of the triazole ring of the adjacent molecule (Figure 3a). The only exception to this pattern was compound **6d**—in the crystal packing of **6d** the OH⋯O supported chains are observed, and such change probably allowed the formation of additional halogen bonding interaction, as indicated in Figure 3b and Appendix A (*ESI*). The intermolecular interactions of derivatives containing fluorine atoms always include CH⋯F contacts; the shortest one, which is observed in **6c,** exhibits the donor-acceptor distance below 3 Å, implicating quite strong interaction. As shown in Figure 3a, the intermolecular forces are usually strongest in the approximate plane of each molecule, which leads to the formation of hydrogen-bonded 2-D arrangements.

Based on the determined crystal structures of compounds **6a**–**e,** the Hirshfeld surfaces (HSs) were mapped with a d_norm_ function, which are illustrated in Figure 4 with fingerprint decomposition. The calculated HSs of molecules **6a**–**e** indicated the presence of strong O–H⋯N contacts between molecules, which are represented by the red spots shown in Figure 4. Moreover, the blue regions correspond to weak interactions, such as C–H⋯H contacts. According to the decomposed fingerprint plots of compounds **6a**–**e**, it was noticed that the most important interactions between molecules are van der Waals forces. The diagram of **6a** shows that the C⋯H (33.4%) bonds are major factors in the crystal packing with hydrophobic H⋯H (24.7%) interactions, which are the next highest contribution. The decomposed fingerprint plots confirmed that with the increase in the number of fluorine atoms in the molecule of **6b** and **6c**, the F⋯X interactions become very important (28.4% in **6b**) or even prevailing (38.7% in **6c**) intermolecular forces. Based on the fingerprint plots of **6d**, it was detected that the major factors in the crystal packing are hydrophobic H⋯H (22.2%) and C⋯H (34.7%) bonds. As in **6a**–**6c** series, with the increase in the number of halogen atoms, their intermolecular interactions grow in strength and become an important structure-building factor. Accordingly, in **6e,** Cl⋯X (30.1%) becomes comparable with the hydrophobic H⋯H plus C⋯H (35.1%).

### 2.3. Computational Studies

#### 2.3.1. The Lipinski’s Rule of Five Calculations

The free access web tool SwissADME server (Swiss Institute of Bioinformatics, Lausanne, Switzerland) was used to predict drug-like physicochemical (PC) properties of compounds **6a**–**e** based on Lipinski’s Rule of Five (molecular weight less than 500, log P or coefficient partition between −5 and 5, less than five HB donors, and less than ten HB acceptors) [8]. Our calculations indicated that compounds **6a**–**e** demonstrate desired drug-like PC features (the collected data are summarized in Table 1).

#### 2.3.2. Molecular Docking

Our previous research indicated that some of the sulfamoylated derivatives of compounds **6a**–**e** (e.g., *MD77* **4**) demonstrated very high STS inhibitory properties, and therefore, they might be recognized as drug candidates in the treatment of hormone-dependent types of cancers. In the present studies, we performed molecular docking calculations to the active sites of several molecular targets (AROM, STS, 17β-HSD1, ERα, and ERβ) for compounds **6a**–**e** using AutoDock Vina 1.1.2 software (Molecular Graphics Laboratory, The Scripps Research Institute, LaJolla, CA, USA). The obtained data indicated that the 4-(1-phenyl-1*H*-1,2,3-triazol-4-yl)phenol core may be useful in the development of novel inhibitors of the abovementioned proteins. The summarized results of docking calculations are presented in Table 2. Subsequently, the visualizations of the examples of docked compounds using VMD 1.9 software (University of Illinois at Urbana—Champaign, Urbana, IL, USA) and detailed identification of plausible interactions using BIOVIA software (Dassault Systémes, Discovery Studio Visualiser, San Diego, CA, USA) were performed.

##### AROM

The calculated binding free energies of compounds **6a**–**d** to AROM were comparable and in the range of −7.5 to −8.3 kcal mol^−1^ (Table 2). Only the binding free energy for compound **6e** was significantly higher (−5.5 kcal mol^−1^). In general, the binding free energies calculated for fluorinated derivatives were slightly better than their chlorinated analogs. However, all of the calculated values were less favorable than the binding free energy of androstenedione (−12.4 kcal mol^−1^), indicating a slightly worse match to the enzyme’s active site than the natural substrate. Analysis of docking modes of compounds **6a**–**e** indicated that triazole rings are close to the heme Fe^2+^ (e.g., 4.14 Å for compound **6b**, Figure 5) analogously as a methyl group of androstenedione (3.94 Å). Furthermore, the halogenated rings of compounds **6a**–**e** occupied the same region of the enzyme’s binding site like a five-membered ring of androstenedione. For example, the distance between one of the fluorine atoms of compound **6b** and Asp309 was 2.36 Å (in the case of androstenedione, the distance between the carbonyl group and Asp309 was 3.02 Å). A more detailed list of the plausible interactions between compounds **6a**–**e** and the AROM enzyme was obtained using BIOVIA and presented in Appendix A (*ESI*).

##### STS

Initially, the docking calculations to the STS protein were performed for compounds **6a**–**e** and E1 (used as a reference). The calculated binding free energies of compounds **6a**–**e** were in the range of −8.1 to −8.3 kcal mol^−1^ (Table 2) and were slightly less favorable than the binding free energy value of E1 (−8.9 kcal mol^−1^). There were no significant differences between binding free energy values for fluorinated and chlorinated analogs. Analysis of docking modes of compounds **6a**–**e** and E1 indicated their similar binding manner to the STS active site. As it is presented in Figure 6A for representative **6e** derivative, we found that the –OH groups of compound **6e** and E1 are in a short distance to the catalytic amino acid residue fGly75 (3.13 and 2.72 Å, respectively) coordinated to the Ca^2+^ ion. Furthermore, we detected that the halogen atoms of compounds **6a**–**e** occupied the same region of the STS active site as the carbonyl group of E1, indicating the presence of possible interactions with Arg98.

However, taking into consideration the mechanism of the enzymatic reaction catalyzed by the STS, we prepared and docked a set of sulfated derivatives of compounds **6a**–**e**. Their calculated binding free energies were in the range of −6.6 to −7.8 kcal mol^−1^ (Table 2) and were more favorable than the binding free energy value of E1S (−6.3 kcal mol^−1^), used as a reference. Figure 6B shows the structure of the sulfated derivative of **6c** and E1S docked into the active site of STS. We found that the sulfate groups of the sulfated derivative of **6c** and E1S were located very close to the fGly75 residue (3.32 and 2.99 Å, respectively). Analogously, as it was detected in the case of phenolic derivatives **6a**–**e**, the halogen atoms of their sulfated derivatives were also in close distances to the Arg98 residue indicating the presence of interactions, which may stabilize the enzyme–ligand complexes.

Importantly, more favorable binding free energy values were calculated for sulfates of compounds **6a**–**e** in comparison with E1S, indicating better binding of ligands based on the 4-(1-phenyl-1*H*-1,2,3-triazol-4-yl)phenol core before catalysis than the natural substrate. On the other hand, less favorable binding free energies were calculated for phenolic derivatives **6a**–**e** in comparison with E1, indicating easier dissociation of the compounds after catalysis. Both of the mentioned observations suggest that the enzymatic reaction for compounds containing 4-(1-phenyl-1*H*-1,2,3-triazol-4-yl)phenol core may be more effective in comparison with the hydrolysis of E1S to E1. A more detailed list of the plausible interactions between compounds **6a**–**e** in their phenolic and sulfate forms and STS enzyme was obtained using BIOVIA and presented in Appendix A (*ESI*).

##### 17β-HSD1

The calculated binding free energies of compounds **6a**–**e** to 17β-HSD1 were comparable (in the range of −7.9 to −8.5 kcal mol^−1^ Table 2) and slightly higher than the binding free energy value of E1 (−8.9 kcal mol^−1^). Fluorinated and chlorinated derivatives both demonstrated similar binding free energy values. Compounds **6a**–**e** docked to the 17β-HSD1’s active site analogously as it was determined for E1 (the halogenated rings of compound **6a**–**e** occupied the same region as a five-membered ring of E1). As it is presented for compound **6c** (Figure 7), we found that the fluorine atoms present in the meta and para positions of the terminal aromatic ring were in close distances to the Ser142 and Tyr155 amino acid residues (the distances of meta-F–HO-Ser142, meta-F–HO-Tyr155 and para-F–HO-Tyr155 were 3.13, 2.95 and 3.31 Å, respectively). In comparison, the distances between the carbonyl group of the E1 and Ser142 and Tyr155 were 5.29 and 4.45 Å, respectively. The –OH groups of compound **6c** and E1 occupied strictly the same region of the enzyme’s active site and were in short distances to the His221 and Gly282 amino acid residues (2.97 and 3.06 Å for **6c**). All plausible interactions between compounds **6a**–**e** and the 17β-HSD1 enzyme were obtained using BIOVIA and collected in Appendix A (*ESI*).

##### ERα and ERβ

The calculated binding free energies of compounds **6a**–**e** to ERα and ERβ were similar (in the range of −7.9 to −8.8 kcal mol^−1^ and −7.6 to −8.9 kcal mol^−1^, respectively, Table 2), suggesting their effective association to the ERα and ERβ binding sites (the binding free energy values of E2 were −10.7 and −11.1 kcal mol^−1^, respectively). In both cases, the binding free energy values calculated for fluorinated derivatives were slightly better than their chlorinated analogs; however, in our opinion, the differences are too low to indicate them as significant. Compounds **6a**–**e** docked to the ERα’s and ERβ’s binding sites in a similar manner as reported for E2. As it is presented in Figure 8A, we found that the –OH groups of compound **6c** and E2 were in short distances to the Glu353 and Arg394 amino acid residues of ERα (2.70 and 2.80 Å for **6c**, respectively; 2.92 and 3.21 Å for E2, respectively). On the opposite side of the binding region of ERα, we detected that the halogens atoms of compounds **6a**–**e** occupied the same region as the -OH group of E2, indicating the presence of interactions with His524. For example, the distance between the para-substituted fluorine atom of **6c** and the nitrogen atom of His524 was 2.72 Å (analogously, the distance between the –OH group of E2 and His524 was 2.93 Å). In case of docking to ERβ’s binding site (Figure 8B), we detected that –OH groups of compounds **6a**–**e** were in short distances to the Glu305 and Arg346 amino acid residues analogously to E2 (for example, 2.90 and 3.63 Å for **6c**, respectively; 2.93 and 2.91 Å for E2, respectively). Furthermore, halogen atoms of compounds **6a**–**e** occupied the same region of ERβ like –OH group of E2. The para-substituted fluorine atom of compound **6c** was at a distance of 3.05 Å to the nitrogen atom of His475 (for comparison, the distance between the –OH group of E2 and His475 was 3.17 Å). Appendix A (*ESI*) summarizes all of the plausible interactions between compounds **6a**–**e** and both ERs detected using BIOVIA.

## 3. Materials and Methods

### 3.1. Synthesis

4-iodophenol, trimethylsilylacetylene, PdCl_2_, PPh_3_, CuI, NEt_3_, 1 M solution of TBAF in THF, sodium ascorbate, CuSO_4_·5H_2_O, *t*-BuONO, TMSN_3_, all of the used aniline derivatives, and solvents are commercially available from Merck (Merck KGaA, Darmstadt, Germany). Solvents were dried and distilled using standard procedures. Melting points (uncorrected) were determined with a Stuart Scientific SMP30 apparatus (Stuart, Stone, UK). NMR spectra were recorded on a Bruker Avance III HD 400 MHz spectrometer (Bruker, Billerica, MA, USA). Chemical shifts are reported in ppm relative to the residue solvent peak (DMSO-d_6_ 2.49 ppm for ^1^H, 39.5 ppm for ^13^C). Coupling constants are given in Hertz. IR spectra were recorded on a Nicolet 8700 spectrometer (Thermo Fisher, Waltham, MA, USA). Mass spectra were recorded on an Agilent 6540 Accurate Mass Q-TOF LC/MS System (Agilent, Santa Clara, CA, USA). Column chromatography was performed using silica gel 60 (230-400 mesh, Merck KGaA, Darmstadt, Germany). Elemental analysis was performed using a CHNS-Carlo Erba EA-1108. Preparative thin-layer chromatography was performed with Polygram SIL G/UV_254_ silica gel (Macherey–Nagel GmbH and Co. KG, Düren, Germany). The detailed procedure for the synthesis of compounds **6a**–**e** was previously described (with the characterization of compounds **6a-c**) [5]. Recrystallization in acetonitrile (ACN) (**6c**–**e**), methanol (**6b**), or acetone (**6a**) allowed to obtain crystals of compounds suitable for X-ray diffraction measurement.

*4-[1-(3-chlorophenyl)-1H-1,2,3-triazol-4-yl]-phenol* **6d**. Yield 70%; mp 208–209 °C; *ν*_max_ (KBr)/cm^−1^ 3458, 1616, 1591, 1466, 1222, 1175, 1040, 839, 681; ^1^H NMR δ_H_ (400 MHz, DMSO) 9.70 (1H, s, OH), 9.19 (1H, s, CH), 8.07 (1H, t, *J* 2.0 Hz, Ar-H), 7.98–7.93 (1H, m, Ar-H), 7.75 (2H, d, *J* 8.7 Hz, Ar-H), 7.66 (1H, t, *J* 8.1 Hz, Ar-H), 7.60–7.55 (1H, m, Ar-H), 6.89 (2H, d, *J* 8.7 Hz, Ar-H); ^13^C NMR δ_C_ (101 MHz, DMSO) 158.2, 148.3, 138.2, 134.7, 132.1, 128.8, 127.3, 121.4, 120.1, 118.9, 118.7, 116.2. Anal. calcd for: C_14_H_10_ClN_3_O: C, 61.89; H 3.71; N, 15.47. Found: C, 61.97; H, 3.60; N, 15.51%. HRMS (*m/z*) [M−H]^−^ calcd 270.0434, found 270.0547.

*4-[1-(3,5-dichlorophenyl)-1H-1,2,3-triazol-4-yl]-phenol* **6e**. Yield 54%; mp 241–244 °C; *ν*_max_ (KBr)/cm^−1^ 3126, 1614, 1591, 1471, 1226, 1177, 1057, 841, 662; ^1^H NMR δ_H_ (400 MHz, DMSO) 9.72 (1H, s, OH), 9.24 (1H, s, CH), 8.08 (2H, d, *J* 1.8 Hz, Ar-H), 7.76 (1H, t, *J* 1.8 Hz, Ar-H), 7.73 (2H, d, *J* 8.7 Hz, Ar-H), 6.89 (2H, d, *J* 8.7 Hz, Ar-H); ^13^C NMR δ_C_ (101 MHz, DMSO) 158.3, 148.4, 138.8, 135.7, 128.3, 127.3, 121.2, 118.9, 116.3. Anal. calcd for: C_14_H_9_Cl_2_N_3_O: C, 54.92; H 2.96; N, 13.73. Found: C, 54.85; H, 2.91; N, 13.86%. HRMS (*m/z*) [M−H]^−^ calcd 304.0044, found 304.0156.

### 3.2. X-ray Diffraction Measurement

Single crystal X-ray diffraction data of the compounds **6a**–**e** were collected at 120(2) K on a Stoe IPDS-2T diffractometer with graphite-monochromated Mo-Kα radiation. Crystals were cooled using a Cryostream 800 open flow nitrogen cryostat (Oxford Cryosystems, Long Hanborough, Oxford, UK). Data collection and image processing were performed with X-Area 1.75 (STOE and Cie Gmbh, Darmstadt, Germany) [9]. Intensity data were scaled with LANA (part of X-Area) in order to minimize differences of intensities of symmetry-equivalent reflections (multi-scan method). Structures were solved by direct methods, and all non-hydrogen atoms were refined with anisotropic displacement parameters by a full-matrix least squares procedure based on F2 using the SHELX–2014 program package [10,11]. The Olex [12] and Wingx [13] suites were used to prepare the final version of the CIF files. Mercury [14] was used to prepare the figures and to calculate the planes of six-membered rings. Hydrogen atoms were refined using an isotropic model with Uiso(H) values fixed to be 1.2 times U_eq_ for –CH groups. The hydrogens of OH groups were refined freely. A summary of crystallographic data is shown in Table 3.

CCDC 2,063,930–2,063,934 contain the supplementary crystallographic data for this paper. These data can be obtained free of charge from The Cambridge Crystallographic Data Centre.

The Hirshfeld surfaces and 2D fingerprint plots were generated using Crystal Explorer 17.5. Crystal structures were imported from CIF files. Hirshfeld surfaces were received using a high surface resolution and mapped with the d_norm_ function.

### 3.3. Computational Studies

#### 3.3.1. The Lipinski’s Rule of Five Calculations

Calculation of the standard properties of compound **6a**–**e** based on Lipinski’s Rule of Five was performed using the SwissADME server (Swiss Institute of Bioinformatics, Lausanne, Switzerland).

#### 3.3.2. Ligands Preparation for Molecular Docking

The 3D structure of ligands **6a**–**e**, their sulfated analogs, and reference compounds (androstenedione, E1S, E1, and E2) were prepared with the Portable HyperChem 8.0.7 Release (Hypercube, Inc., Gainesville, FL, USA). Prior to docking calculations, the structure of each ligand was optimized using an MM+ force field and the Polak–Ribière conjugate gradient algorithm (terminating at a gradient of 0.05 kcal mol^−1^ Å^−1^).

#### 3.3.3. Protein Preparation for Molecular Docking

The X-ray structures of the AROM, STS, 17β-HSD1, ERα and ERβ used for molecular modeling studies were taken from the Protein Databank (Protein Data Bank accession codes: 3EQM, 1P49, 6MNC, 1A52, and 5TOA, respectively). After standard preparation procedures (including removal of water molecules and other ligands as well as addition hydrogen atoms and Gasteiger charges to each atom), docking analysis was carried out.

#### 3.3.4. Molecular Docking

Docking studies were carried out using Autodock Vina 1.1.2 software (The Molecular Graphic Laboratory, The Scripps Research Institute, La Jolla, CA, USA) [15]. For the docking studies, the corresponding grid box parameters were used:

- AROM: a grid box size of 24 Å × 24 Å × 24 Å centered on Asp309 amino acid residue (x = 88.230, y = 49.522, z = 51.205);

- STS: a grid box size of 24 Å × 24 Å × 24 Å centered on the fGly75 amino acid residue (x = 72.135, y = −1.720, z = 28.464);

- 17β-HSD1: a grid box size of 24 Å × 24 Å × 24 Å centered on Ser142 amino acid residue (x = 20.760, y = −6.625, z = −22.837);

- ERα: a grid box size of 24 Å × 24 Å × 24 Å centered on Glu353 (x = 105.702, y = 19.417, z = 103.747);

- ERβ: a grid box size of 24 Å × 24 Å × 24 Å centered on Met336 (x = 16.635, y = 41.368, z = 18.491).

Graphic visualizations of the 3D model were generated using VMD 1.9 software (University of Illinois at Urbana—Champaign, Urbana, IL, USA). Identification of the ligand–protein interactions was performed using Discovery Studio Visualiser v20. 1. 0. 19,295 (BIOVIA, Dassault Systémes, San Diego, CA, USA).

## 4. Conclusions

In summary, we synthesized five 4-(1-phenyl-1*H*-1,2,3-triazol-4-yl)phenol derivatives containing halogen atoms, **6a**–**e**. Single crystals of such compounds were obtained, and their crystal structures were successfully confirmed. The determined crystal structures were used to calculate the Hirshfeld surfaces for each of the synthesized compounds, which indicated that the structures of all compounds are mainly characterized by H⋯H, C⋯H, and X⋯H (where X is F or Cl) interactions. The decomposed fingerprint plots confirmed that with the increase in the number of halogen atoms in the molecules, the interactions involving fluorine or chlorine atoms become very important or even prevail the intermolecular structure-building factor. Furthermore, the performed docking studies proved potentially effective binding of compounds based on a 4-(1-phenyl-1*H*-1,2,3-triazol-4-yl)phenol core to the active sites of several molecular targets (AROM, STS, 17β-HSD1, ERα, and ERβ). Importantly, we also noticed that the halogen atoms of compounds **6a**–**e** play an important role in the putative binding process, and their presence is crucial for the creation of stabilizing interactions in the active sites of examined molecular targets. This finding corresponds well with the crystallographic data and Hirshfeld surfaces analysis, which indicated that the interactions involving halogen atoms are a very important structure-building factor.

Our research indicated that the studied compounds might be a great starting point in the development of inhibitors of enzymes involved in the hormone biosynthesis and signaling pathways. Importantly, the calculated binding free energies for sulfates of compounds **6a**–**e** were better than the binding free energy of the STS native substrate (E1S). As we mentioned, we previously reported that sulfamoylated derivatives of halogenated 4-(1-phenyl-1*H*-1,2,3-triazol-4-yl)phenols are potent STS inhibitors, which proves our present findings. It is also worth noting that the binding free energies for compounds **6a**–**e** were comparable to the binding free energy of 17β-HSD1 native substrate (E1), indicating that such compounds may also potentially demonstrate 17β-HSD1 inhibitory properties. However, such hypothesis needs to be evaluated in further biological experiments. Additionally, our docking studies and calculations of standard PC properties indicated that compounds **6a**–**e** demonstrate favorable drug-like features and may be potentially used as drugs. However, due to the limited accuracy of the utilized AutoDock Vina, further computational studies using more accurate methods (e.g., free energy perturbation calculations) should be performed for finally designed compounds.

## Data Availability

Not applicable.

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
