# Peer review of "Novel 1,2,3-Triazole Derivatives as Mimics of Steroidal System—Synthesis, Crystal Structures Determination, Hirshfeld Surfaces Analysis and Molecular Docking"

_molecules, 2021, doi:10.3390/molecules26134059_

Round 1

Reviewer 1 Report

This resubmitted manuscript can't be accepted in the current form and I will justify this opinion below.

Previously, I have asked the Authors to improve the level of novelty. I have provided specific instructions how to do this. In particular, I have requested including the other receptor (ER Beta) which the Authors have done, but more importantly I have asked for free energy perturbation (FEP) calculations based on molecular dynamics simulations. The Authors have not done this. Instead, they focus solely on the not very accurate results from Auto Dock Vina. Those deltaG from Vina are based on “static” molecular mechanics calculations and thus their accuracy is questionable.

I totally agree with the Reviewer 2 who has pointed out the low level of novelty of this work which is just a “supplement” to the other work, published previously by the same Authors.

Previously, I have suggested one of the easiest ways to somehow improve the level of novelty but the Authors have not taken this suggestion, I don’t know why. Either they didn’t know how to do this (as some of the software for both MD and FEP are free and therefor the argument that they don’t have a license would be invalid) or they do not agree with my opinion.

Concluding, I think that the level of novelty of this work should be greatly improved.

Author Response

Dear Sir/Madam,

we gratefully acknowledge you for the review. We agree with the statement that the results of free energy perturbation (FEP) calculations could be more accurate than those obtained using Autodock Vina. However, the performance of FEP calculations is very computationally expensive and time-consuming (taking into consideration all of our compounds including all of examined molecular targets). We consulted the possibility of the performance of such studies with three experts in the field of computational chemistry and in our case, appropriate calculations for all of our compounds including all of the examined molecular targets may take several months. Therefore, in our opinion, they are not suitable at the step of drug design process presented in the manuscript. As we previously mentioned, we did not state that described compounds are inhibitors of all considered molecular targets, but we indicated that their core may be used as perfect starting point in the development of novel bioactive agents by other research groups, what was theoretically confirmed by our docking studies. However, their structures may be (and should be) modified in order to develop more specific agents (e.g., by the addition of the sulfamate moiety in case of the development of STS inhibitors). Therefore, in our opinion the accuracy achieved with Autodock Vina is sufficient at this step of studies, however, more accurate FEP calculations should be performed for finally designed compounds (e.g., sulfamates). We appreciate your suggestion in this matter, and for sure, we will include such calculations in our further studies dedicated to specific STS inhibitors, which are our main area of interest. According to your comment we have added the sentence, which indicate that the results of Autodock Vina are just predictions with limited accuracy and they should be confirmed by more accurate computational methods (e.g., FEP calculations) for finally designed compounds.

One more time, thank you for your review. We hope that our explanation will convince you that our revised manuscript is suitable for publication in Molecules. Please take note, that in the present paper we described not only the molecular docking studies for our compounds but we also presented the way for synthesis and obtaining them in the monocrystalline forms. Furthermore, the most important, we described the crystallographic data for all of the synthesized compounds, which was not previously reported. Next, the determined crystal structures were used to calculate the Hirshfeld surfaces for each of the synthesized compounds. Both, X-ray data and Hirshfeld surfaces analysis, may be potentially helpful in the identification of the intermolecular interactions that can arise between the studied molecules and their environment, e.g., enzymes and receptors. For this reason, in our opinion, the described results may be crucial in the development process of bioactive compounds performed by us and other research groups.

Yours faithfully,

Mateusz Daśko

Reviewer 2 Report

I recommend acceptance of the paper

Author Response

Dear Sir/Madame,

we are happy to know that our manuscript meets all of your requirements and we appreciate your recommendation of acceptance.

Yours faithfully,

Mateusz Daśko

Round 2

Reviewer 1 Report

I appreciate the honest answer of the Authors and their efforts made in order to improve the quality of the manuscript.